# Docosahexaenoic Acid as Master Regulator of Cellular Antioxidant Defenses: A Systematic Review

**DOI:** 10.3390/antiox12061283

**Published:** 2023-06-15

**Authors:** Sara Margherita Borgonovi, Stefania Iametti, Mattia Di Nunzio

**Affiliations:** Department of Food, Environmental and Nutritional Sciences (DeFENS), University of Milan, Via Celoria 2, 20133 Milan, Italy; sara.borgonovi@unimi.it (S.M.B.); stefania.iametti@unimi.it (S.I.)

**Keywords:** docosahexaenoic acid, antioxidants, Nrf2

## Abstract

Docosahexaenoic acid (DHA) is a polyunsaturated fatty acid that benefits the prevention of chronic diseases. Due to its high unsaturation, DHA is vulnerable to free radical oxidation, resulting in several unfavorable effects, including producing hazardous metabolites. However, in vitro and in vivo investigations suggest that the relationship between the chemical structure of DHA and its susceptibility to oxidation may not be as clear-cut as previously thought. Organisms have developed a balanced system of antioxidants to counteract the overproduction of oxidants, and the nuclear factor erythroid 2-related factor 2 (Nrf2) is the key transcription factor identified for transmitting the inducer signal to the antioxidant response element. Thus, DHA might preserve the cellular redox status promoting the transcriptional regulation of cellular antioxidants through Nrf2 activation. Here, we systematically summarize the research on the possible role of DHA in controlling cellular antioxidant enzymes. After the screening process, 43 records were selected and included in this review. Specifically, 29 studies related to the effects of DHA in cell cultures and 15 studies concerned the effects of consumption or treatment with DHA in animal. Despite DHA’s promising and encouraging effects at modulating the cellular antioxidant response in vitro/in vivo, some differences observed among the reviewed studies may be accounted for by the different experimental conditions adopted, including the time of supplementation/treatment, DHA concentration, and cell culture/tissue model. Moreover, this review offers potential molecular explanations for how DHA controls cellular antioxidant defenses, including involvement of transcription factors and the redox signaling pathway.

## 1. Introduction

Docosahexaenoic acid (DHA) is a highly polyunsaturated fatty acid (PUFA) of the n-3 series with 22 carbon atoms and 6 cis double bonds. DHA plays a crucial role in lipid metabolism [1], in membrane structure [2], in cell signaling [3], and in inflammation [4]. Epidemiological studies have demonstrated that diets rich in DHA have a positive effect against several types of disease [5,6,7]. Marine-based fish and fish oil are the most popular and well-known sources of DHA.

Plasma non-esterified DHA derived from chilomicrons and VLDLs enters the cells via passive diffusion or transporters such as fatty acid transport protein or fatty acid transporter CD36 [8]. Inside the cells, non-esterified DHA is converted by acylCoA synthases to DHA-CoAs, which are substrates for β-oxidation, desaturation/elongation and assimilation into complex lipids, i.e., phospholipids in the plasma membrane [9].

The physicochemical properties of the membrane bilayer and the chemical reactivity of the fatty acids that compose the membrane are two inherent traits of the membrane phospholipids that regulate their fluidity and determine their susceptibility to oxidative damage. The first property is related to the fact that oxygen and reactive species are more soluble in the fluid lipid bilayer than in the aqueous solution. Consequently, membrane lipids become primary targets of oxidative damage. The second and more significant property is related to the fact that PUFA residues of phospholipids are extremely sensitive to oxidation [10]. PUFAs are usually oxidized by a well-known mechanism called “free radical oxidation”. This theory involves an attack of oxygen at the allylic position with the formation of unsaturated hydroperoxides. These hydroperoxides also take part in the auto-oxidation and thus initiate a chain reaction [11].

Due to its high unsaturation, DHA susceptibility to free radical oxidation may represent the other side of the coin. This uncontrolled oxidation of DHA may have a variety of metabolic and physiological repercussions, such as altering the lipid bilayer’s structure and function [12] or producing harmful byproducts such malondialdehyde and alkenals [13]. Therefore, a theoretical concern remains on using DHA for preventing chronic diseases whenever oxidative stress is one of the underlying mechanisms.

The human body implemented several strategies to counteract the effects of excess free radicals based on antioxidant molecules [14]. Endogenous antioxidants, which are products of the body’s metabolism, may be enzymatic or non-enzymatic. Enzymatic antioxidants playing an essential role in the first line of defense are superoxide dismutase (SOD), catalase (CAT), glutathione peroxidases (GPx), and peroxiredoxins (PRx) [15,16]. The second line of defense involves non-enzymatic antioxidants such as glutathione (GSH) and thioredoxin (TRx), characterized by the ability to rapidly inactivate ROS and oxidants [17].

This articulated mechanism is regulated at the cellular level through a cis-acting element called antioxidant or electrophile response elements (ARE/EpRE) [18]. Nuclear factor erythroid 2-related factor 2 (Nrf2) is the key transcription factor for transmitting the inducer signal to AREs, and many food bioactive compounds were identified as Nrf2 inducers [18]. As DHA has been shown to modulate transcription of genes related to lipid metabolism, such as stearoyl-Coenzyme A desaturase 2 and 3-hydroxy-3-methyl-glutaryl-coenzyme A reductase [19,20], by interacting with several nuclear receptors, such as peroxisome proliferator-activated receptor (PPAR) and sterol regulatory element binding protein (SREBP) [21,22,23], it is reasonable to assume that DHA could maintain the cellular redox status promoting the transcriptional regulation of antioxidant expression through Nrf2 activation.

In light of these considerations, we tried to overview studies on the potential effect of DHA in regulating cellular antioxidant enzymes. This review highlights DHA’s health-related potential and hypothesizes possible molecular scenarios between DHA and Nrf2 in regulating cellular antioxidant defenses.

## 2. Methods

This systematic review was performed according to the Preferred Reporting Items for Systematic Reviews and Meta-Analyses guidelines (PRISMA) [24]. The protocol for this systematic review was registered on INPLASY (INPLASY202360017) and is available in full at inplasy.com (https://inplasy.com/inplasy-2023-6-0017/). The search was carried out by using the PubMed database in December 2022, and was conducted using the following keywords and Boolean operators: “docosahexaenoic acid” OR “DHA” OR “C22:6” AND “antioxidant” NOT “review”. The initial search yielded 1063 hits. During the screening process (reviewing titles), 941 records were excluded. After abstract analysis, another 84 articles were ousted. Altogether, 43 records were selected and included in this review. Specifically, 29 studies related to the effects of DHA in cell cultures and 15 studies concerning the effects of consumption or treatment with DHA in animal models were deeply analyzed. One study concerned cell culture and animal models and was allocated to each group. Chosen studies were published between 1998 and 2021 without restriction regarding period or publication status. Exclusion criteria were: (i) titles irrelevant to the research topic; (ii) abstract inappropriate or not related to the research topic; (iii) studies that used n-3 PUFA rich oils which would not allow us to discriminate the effect of DHA from other n-3 PUFAs; (iv) studies that co-administrated DHA with other compounds; (v) studies that used DHA oxidation products to better reflect normal nutritional conditions; and (vi) studies or data with inadequate statistical analysis or inappropriate controls. Reviews, letters, abstracts, and articles without a complete text in the English language were also excluded. Two independent investigators (S.M.B and M.D.N.) checked the titles and abstracts of the studies, and disagreements between the two reviewers were resolved through a mediator (S.I.). Primary outcomes include the most relevant variables to answer the research question (the modulation of cellular antioxidant defenses by DHA), while secondary outcomes include additional variables to help the interpretation of the results of the primary outcomes. The detailed selection process is presented in Figure 1.

## 3. Results and Discussion

### 3.1. Effect of DHA Supplementation on Antioxidant Defenses in Cultured Cells

Although studies on humans remain the gold standard for evaluating the relationship between nutrients and health, the development of reliable in vitro/ex vivo models allow the investigation of the cellular/molecular mechanisms and represents a first—and undoubtedly necessary—step when investigating the health-promoting properties of food components [25]. Table 1 summarizes the data published on the effect of DHA supplementation on antioxidant defenses in cultured cells (Table 1).

The number of available studies is limited to 29 cases. Eight of them used primary cells, while twenty-one relied on cell lines. Primary cells included human fibroblast [28] and peripheral blood mononuclear cells [26], bovine endothelial cells [27], carp brain cells [31], and rat thymocytes [26], hepatocytes [29], hippocampal neurons [30], and astrocytes [32,33]. Among cell culture studies, six were conducted on hepatocytes [38,39,40,41,42,43], four on nervous system cells [44,45,46,47], four in adrenal cells [48,49,50,51], two on pancreatic [52,53] and breast cells [37], and only one study was conducted on ovarian [34], skeletal muscle [35], adipocyte [36], and monocyte cells [54].

The studies listed above cover a wide variety of the effects of DHA supplementation on antioxidant defense systems. However, comparing the reported results is difficult because essential experimental conditions varied significantly between individual labs. Among the studies, the DHA concentration spanned a wide range of concentrations (from approximately 0.01 µM [46] to 150 µM [34]) as did supplementation times (from 1 h [26,33,34] to 10 days [27]). In addition, 10 studies were conducted in basal conditions [27,28,34,35,38,39,41,44,45,52], 8 studies with simultaneous or subsequent exogenous stress [26,30,40,42,43,48,49,51], and 11 studies included both conditions [29,31,32,33,36,37,46,47,50,53,54]. Among them, 20 studies measured the enzymatic activity [26,29,30,31,32,33,35,37,38,39,40,41,42,44,45,47,48,49,50,51] while 19 and 8 studies evaluated the antioxidant defenses at the post-transcriptional [27,28,30,31,33,35,37,38,39,40,43,44,45,47,49,50,52,53,54] and transcriptional level, respectively [36,37,41,44,46,47,53,54]. Furthermore, two studies evidenced an increase in cellular oxidative markers [27,45], eight studies found a decrease [26,29,31,42,45,49,50,51], and four studies reported no noticeable effects [29,38,40,50].

Despite these limitations, most of the studies (24) reported how DHA supplementation was able to positively regulate the inducible expression and/or activity of antioxidant species [27,28,29,30,32,33,37,38,39,40,41,42,43,44,45,46,47,48,49,50,51,52,53,54]. Four studies reported no noticeable outcomes [26,31,35,36] and only three reported a negative effect [26,27,34]. Taken together, the results summarized here indicate how DHA may represent an effective promoter of cellular antioxidant defenses. In any case, future studies should investigate a possible inhibitory effect of DHA in modulating antioxidant defenses as reported in some studies.

### 3.2. Effect of DHA Treatment on Antioxidant Defenses in Animal Models

Table 2 summarizes the studies based on the use of in vivo preclinical animal models employed to investigate the possible use of a DHA-rich diet or DHA treatment in the modulation of cellular antioxidant defenses.

The number of studies conducted on animal models is limited to 15 cases. Twelve of them involved rats [55,56,57,58,59,60,61,62,63,64,65,66], whereas three were conducted in mice [43,67,68]. Seven studies were focused on the liver [43,55,56,59,60,61,68], six on the nervous system [58,59,61,62,63,64], three on plasma/serum [55,65,68], and only one study was reported for the kidney [56], retina [57], spleen [66], stomach [67], and erythrocytes [65]. In addition, four studies were carried out in normal conditions [55,56,61,65], four studies in association with injuries [43,59,63,64], and seven studies in both conditions [57,58,60,62,66,67,68].

Again, the DHA treatment spanned a range from 2.5 [59,61] to 1500 mg/d/kg bw [55] or a diet containing DHA from 5 [56] to 10 g/kg diet [65], with a time of somministration ranging from 1 day (single administration) [64,67] to 90 days [62].

Among them, 13 studies measured the enzymatic activity [43,55,57,58,59,60,61,62,63,64,65,67,68], while 9 and 2 studies evaluated the antioxidant defenses at the post-transcriptional [43,55,56,57,58,60,63,67,68] and transcriptional level, respectively [55,66]. Furthermore, one study evidenced an increase in cellular oxidative markers [56], four studies found a decrease [57,63,64,67], and nine studies reported no noticeable effects [43,55,56,57,58,62,65,67,68].

Despite these limitations, 14 studies reported a promoting effect of DHA supplementation on at least one expression or activity of antioxidant molecules [43,56,57,58,59,60,61,62,63,64,65,66,67,68], whereas only an individual study reported no effects [55].

### 3.3. Role of DHA in the Intracellular Redox Homeostasis Mechanism

To maintain their energy metabolism, mammalian cells have evolved to use oxygen as a final electron acceptor. Consequently, they must deal with a collection of undesired oxygenated byproducts produced due to these oxygen-dependent metabolic processes. These oxygenated byproducts are collectively referred to as ROS [69]. At low levels, ROS can undergo reactions with biological macromolecules contributing to redox signaling and biological function [69], but—at supraphysiological concentrations—they may undergo aspecific reactions that generate other reactive species with potentially toxic consequences [70]. 

The susceptibility of fatty acids to oxidation is thought to depend directly on their degree of unsaturation. Reportedly, the oxidation rate of a fatty acid or its esters is typically increased by at least one or two factors for every additional double bond in a fatty acid [71], thus placing DHA in the highest ranks among oxidizable species. However, various in vitro [26,31,40,42,49,50,51] and in vivo studies [43,57,58,63,64,67,68] suggest that the relation between the chemical structure of DHA and, even from a theoretical standpoint, its vulnerability to ROS oxidation is not as easy to predict.

Most of the selected in vitro and in vivo studies considered in this review reported a general promoting effect at the transcriptional or post-transcriptional level. Given the evidence for the role of DHA in the development of chronic diseases, it appears necessary to assess the molecular mechanism at the basis of the protective role of DHA.

The cap’n’collar basic-region leucine zipper transcription factor Nrf2, encoded by NFE2L2, is a master regulator of intracellular redox homeostasis because, in response to oxidative stress, it orchestrates induction of a battery of genes such as SOD, NAD(P)H quinone dehydrogenase 1, and heme oxygenase-1 that serve to increase the antioxidant and detoxifying capacity of the cell [72]. This tight control of Nrf2 is achieved by a repressor protein Keap1, a cysteine redox-sensitive factor, that, under normal conditions, serves as a Nrf2-specific adaptor protein for the Cullin-3 ubiquitin ligase complex and perpetually targets Nrf2 in the cytoplasm for degradation by the 26S proteasome [73]. Dissociation of Keap1/Nrf2 complex by oxidants leads to transportation and accumulation of Nrf2 to the nucleus where it binds ARE/EpRE sequences in the promotor region of several genes related to phase II drug conjugation, scavenging of H_2_O_2_, and GSH- and Trx-based antioxidant systems [74]. Since oxidative stress is associated with several diseases [75,76,77] and many of these ailments have been demonstrated to be prevented by DHA [78,79,80], it might not surprising that DHA could reversibly activate Nrf2 and promote induction of cellular antioxidant defenses. DHA itself is not a ligand for Nrf2, so one crucial issue that remains to be determined is the endogenous activation underlying the transcriptional responses elicited by DHA.

Under physiological conditions, DHA can be oxidized enzymatically or non-enzymatically. In the enzymatic oxidation pathway, cyclooxygenase and lipooxygenase catalyze the conversion of DHA to produce a large variety of oxidation metabolites, including hydroperoxide and hydroxide positional isomers [81]. In addition, ROS can oxidize DHA through non-enzymatic reactions that release highly electrophilic species, including neuroprostane and hydroperoxide break-down products such as 4-hydroxy-2-hexenal (HHE) [82,83]. As a result, oxidized DHA, or its derived electrophilic species, may react with Keap1 sulfhydryls, altering Keap1 secondary structure which is followed by a loss of association between Keap1 and Cullin-3. This in turn inhibits Nrf2 ubiquitination, leading to stabilization and nuclear translocation of Nrf2 and to the subsequent induction of Nrf2 target genes [84]. In support of this hypothesis, one recent observation indicates that low but significant levels of HHE are generated upon DHA supplementation [85] just before changes in gene expression are observed. Furthermore, 15-deoxy-Δ12,14-prostagandin [86], F4 neuroprostanes [83], and HHE [87] have been demonstrated to be activators of Nrf2 and to induce expression of cytoprotective enzymes [88].

After incorporation in the plasma membrane, DHA may profoundly influence cellular membrane composition affecting membrane fluidity, phase behavior, permeability, fusion, flip-flop, and protein function [89]. DHA acyl chains have been shown to affect bilayer properties including lateral pressure, microviscosity, curvature, permeability, elasticity, microdomain formation, and hydrophobic match due its high conformational flexibility arising from the low potential energy barriers to rotation about the single carbon–carbon bonds [90]. Moreover, DHA infiltrates rafts and non-raft membrane microdomains, disrupts raft clustering, and increases the size of rafts [91]. The rearrangement of membrane microdomains may have implications for the raft platform signaling and collocation of the transmembrane protein into or out of rafts. Membrane physical properties are mainly affected by lipid composition, and as previous studies have indicated, the activity of G protein-coupled receptors (GPCRs) located in plasma membranes are influenced by the surrounding fluidic membranes [92,93]. Recently, Yoshida et al. demonstrated, using nanodisc technology to control membrane properties, that increased membrane fluidity shifted the equilibrium toward an active form of the receptor through conformational changes [94]. After activation, GPCR transmits signals by downstream pathways leading to the regulation of physiological processes, including antioxidant response [95]. DHA-containing lipids enhance the function of the prototypical GPCR rhodopsin, which simulation studies have explained takes place as a result of the high conformational flexibility of DHA chains [96]. This provides hybrid lipids with a high affinity for the rough surface of GPCRs, further promoting protein–protein interactions.

The Ras/Raf/MAP/ERK kinase (MEK)/ERK cascade couples signals from cell surface receptors to transcription factors [97] involving the MAPK cascade [98]. Nicolini et al. reported that N-Ras prefers a liquid-disordered lipid phase, regardless of the lipid anchor system [99]. Furthermore, several protein kinases such as protein kinase C (PKC), mitogen-activated protein kinase (MAPK), phosphatidylinositol 3-kinase (PI3K), c-Jun N-terminal kinase (JNK), and extracellular-signal-regulated kinase (ERK) have been found to phosphorylate Nrf2 [100,101,102] blocking the KEAP1–Nrf2 interaction and the subsequent KEAP1-dependent proteasomal degradation of Nrf2 [103]. In this complicated but intriguing scenario, DHA-enriched membranes may facilitate Ras binding with the guanine nucleotide exchange factor SOS activating the Ras/Raf/MEK/ERK downstream signaling pathway, causing subsequent phosphorylation of Nrf2 and, thereby, its nuclear accumulation and ARE-driven transcription.

PPARs, including α, δ, and γ isoforms, comprise a subfamily of the nuclear receptor superfamily that is highly expressed in mammalian tissues [104]. Each PPAR subtype is located in the cytoplasm [105] and, after activation by specific ligands, enters to the nucleus heterodimerizing with the retinoid X receptor (RXR) before binding to the PPAR responsive element (PPRE) of specific target genes [106]. Natural products, including DHA and its metabolites, serve as endogenous PPARs ligands, which exert adaptive metabolic responses to changes in metabolic status in various tissues [107,108]. Several studies have shown that PPARs ligands can transcriptionally modulate antioxidants such as TRx [109], GPx [110,111], CAT [112], and SOD [21] due the presence of a PPRE in the promotor regions of the coding gene sequence [113,114]. In addition, evidence indicates that PPARs may be phosphorylated by several kinases such as protein kinase A, MEK/ERK, and p38 kinase [115,116,117,118], all of which affect the A/B domain of the receptor and modulate its ligand-independent AF-1 transactivating function [119]. Several studies also strongly support a reciprocal regulation of the Nrf2 and PPARγ pathways to reinforce the expression of one another [120,121,122,123]. In this sense, Nrf2 and PPARγ pathways seem to be connected by a positive collaborative feedback loop, which maintains the expression of both transcription factors and their target antioxidant genes in a simultaneous manner [124].

## 4. Conclusions

Considering the results obtained in the present review and the extensive search of relevant information available in the literature, we have proposed a scheme to give a logical explanation regarding the potential mechanisms of action of DHA in the modulation of cellular antioxidant defenses (Figure 2).

Despite DHA’s promising and encouraging effects at modulating the cellular antioxidant response, differences observed among the reviewed studies may be accounted for by the different experimental conditions adopted. In fact, unlike the genetically predetermined protein profile, the diet profoundly influences the acyl composition, and several studies have shown a time- and concentration-dependent effect on incorporating DHA into cellular lipids [125,126]. In this line, a very recent paper conducted in football players has demonstrated a dose–response incorporation of DHA into red blood cell membranes up to 6 g·d^−1^, which can be used to rapidly achieve a desired omega-3 index (>8%) in only 8 weeks [127].

In addition, although various cells readily take up DHA, its accumulation is organ-specific, with a higher content in the brain, liver, and heart respecting plasma, pancreas, and erythrocytes [128,129]. At a cellular level, the uptake of long chain fatty acids is mainly regulated by the fatty acid transporter CD36, a transmembrane glycoprotein highly expressed in tissue with high fatty acid uptake [130] with a pivotal role in cellular lipid homeostasis [131].

Moreover, DHA bioavailability depends on the chemical form it conveys. Recent evidence indicates that DHA esterified in phospholipid and triglyceride is more readily absorbed by the body than in ethyl ester form [132]. Taken together, discrepancies in these terms in the studies selected in this review may have determined substantial differences in DHA accumulation and should be deeply considered in future studies to evaluate the minimum effective treatment times and concentrations of DHA according to the cellular models adopted. In any case, further studies including pharmacokinetic/pharmacodynamic modeling and human trials which consider not only DHA in particular but n-3 PUFA in general should be taken into deep consideration.

To conclude, the identification of molecular mechanisms involved in redox metabolism is an important issue for the development of therapies for chronic disorders, and although further studies are needed, the comprehensive vision offered here may help to address future studies toward specific pathways and to provide molecular-based support to any recommendation pertaining the food/health relationship.

## Figures and Tables

**Figure 1 antioxidants-12-01283-f001:**
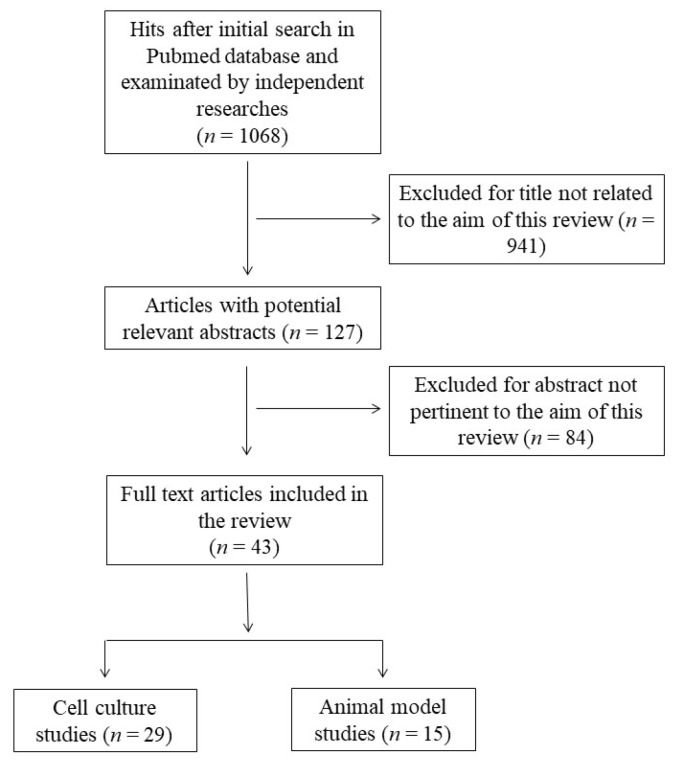
Flow chart of papers included in this review.

**Figure 2 antioxidants-12-01283-f002:**
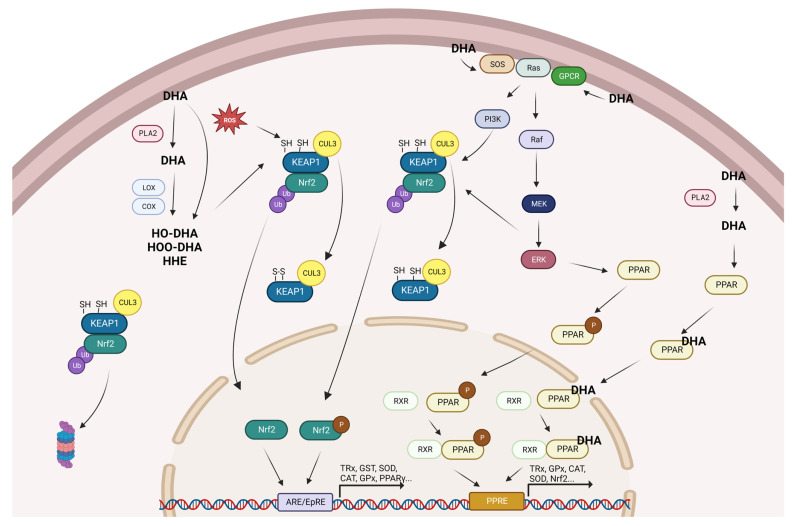
Summary of the proposed mechanisms for the promotion of antioxidant gene expression by DHA. CAT: catalase; Cox: ciclooxygenase; Cul3: cullin 3; DHA: docosahexaenoic acid; ERK: extracellular signal-regulated kinases; GPCR: G protein-coupled receptors; GPx: glutathione peroxidase; GST: glutathione S-transferases; HO-DHA: hydroxy-DHA; HOO-DHA: hydroperoxy-DHA; Keap1: kelch-like ECH-associated protein 1; Lox: lipoxygenase; MEK: mitogen-activated protein kinase kinase; Nrf2: nuclear factor erythroid 2-related factor 2; P: phosphorylated; PI3K: phosphoinositide 3-kinase; PLA2: phospholipase A2; PPAR: peroxisome proliferator-activated receptor; ROS: reactive oxygen species; RXR: retinoid X receptor; SOD: superoxide dismutase; SOS: son of sevenless; and TRx: thioredoxin.

**Table 1 antioxidants-12-01283-t001:** Summary of findings related to the effect of DHA supplementation on cell culture.

Ref.	Cell/Tissue Model	DHA Concentration/Time of Incubation	Exogenous Treatment	Primary Outcomes	Secondary Outcomes
[26]	Primary human peripheral blood mononuclear cells	Pretreatment with 5 µM for 1 h	Subsequent oxidative stress by 0.5, 1, 2, and 5 mM H_2_O_2_ for 10 min	Oxidative condition: ↓ GPx activity at 1 mM H_2_O_2_	Not-oxidative condition: ↑ DHA content in membrane PLs
					Oxidative condition: ↓ MDA content at 1 and 1.5 mM H_2_O_2_; ↑ cytosolic cAMP-PDE at 1, 2, and 5 mM H_2_O; ↑ cytosolic cGMP-PDE content at 1 and 5 mM H_2_O_2_
	Primary rat thymocytes	Pretreatment with 5 µM for 1 h	Subsequent oxidative stress by 0.5, 1, 2, and 5 mM H_2_O_2_ for 10 min	Oxidative condition: ↔ GPx activity	Not-oxidative condition: ↑ DHA content in membrane PLs
					Oxidative condition: ↔ cytosolic cAMP-PDE and cGMP- PDE content
[27]	Primary bovine retinal endothelial cells	10 and 100 µM for 2 cell passages (approximately 10 d)	Not present	↓ cGPx protein expression and activity at 10 µM	↑ MDA content; ↔ αTC and γTC content; ↑ DHA content in PC and PE; ↔ GPx4 activity
	Primary bovine aortic endothelial cells	10 and 100 µM for 2 cell passages (approximately 10 d)	Not present	↑ cGPx protein expression at 100 µM	↑ MDA content; ↔ γTC and ↓ αTC content at 100 µM; ↑ DHA content in PC and PE; ↑ GPx4 activity at 100 µM
[28]	Primary human fibroblasts	5, 15, 30, and 60 µM for 48 h; 30 µM for 4, 8, 24, 48 h, and 7 d	Not present	↑ GSH content at 30 µM and 60 µM for 48 h, and 30 µM for 7 d	↑ γGCL and GST activity at 30 µM and 60 µM for 48 h; ↑ GR activity at 15 µM, 30 µM, and 60 µM for 48 h; ↑ γGCL mRNA expression at 30 µM for 4, 8, 24, and 48 h; ↑ GR mRNA expression at 30 µM for 24 and 48 h; ↑ HO-1 mRNA at 30 µM for 4 and 8 h; ↑ ROS formation at 30 µM for 4, 8, 24, and 48 h; ↑ cell DHA content at 30 µM for 48 h; ↔ lipoperoxides content at 30 µM for 48 h; ↑ γGCL, GR, and GST activity at 30 µM for 7 d
[29]	Primary rat hepatocytes	5, 10, and 20 µM for 48 h	Concomitant oxidative stress by 5 and 10 µM TCDD for 48 h	Basal condition: ↑ TAC	Basal condition: ↔ TOS and 8-OHdG content
				Oxidative condition: ↑ TAC	Oxidative condition: ↓ TOS content at 20 µM DHA with 5 and 10 µM TCDD; ↓ 8-OHdG content
[30]	Primary rat hippocampal neurons	5 µg/mL (15.2 µM), 30 µg/mL (91.3 µM), and 50 µg/mL (152 µM) for 24 h	Concomitant cytotoxic stress by 0.5 mM glutamate for 1 h	Excitotoxic condition: ↑ GPx activity and ↔ GSH content	Excitotoxic condition: ↑ GR activity; ↓ NO formation
[31]	Primary carp brain cells	Pretreatment with 30 nM for 48 h	Subsequent oxidative stress by 1 mg/L (1.39 µM) fullerene (C60) for 2 h	Basal condition: ↔ TAC and GSH content	Basal condition: ↓ ROS and TBARS content; ↔ Cys content
				Oxidative condition: ↔ TAC and GSH content	Oxidative condition: ↓ ROS and TBARS content; ↔ Cys content
[32]	Primary rat astrocytes	5 µM for 24 h	Concomitant oxidative stress by 10 µM UCB for 20 h	Basal condition: ↑ SOD, CAT, and GPx activity	Basal condition: ↔ apoptosis
				Oxidative condition: ↑ SOD, CAT, and GPx activity	Oxidative condition: ↓ apoptosis
[33]	Primary rat astrocytes	Pretreatment with 10, 30, and 50 µM for 1, 2, and 3 h; 30 µM for 24 h	Subsequent oxidative stress by 0.5 mM for 2 h	Basal condition: ↑ GSH content at 30 µM for 24 h	Basal condition: ↓ ROS content at 30 and 50 µM; ↑ Nrf2 protein expression and binding activity; ↑ γGCL and GPx4 expression at 30 µM for 24 h; ↑ DHA content in mitochondrial and plasma membrane PL at 30 µM for 24 h
				Oxidative condition: ↑ GSH content and GPx4 activity at 30 µM for 24 h	Oxidative condition: ↓ ROS content at 30 µM for 1, 2, and 3 h; ↓ and ↑ γGCL and GPx4 expression at 30 µM for 24 h, respectively
[34]	Human ovarian cancer cell line (A2780)	100 µM for 1, 4, and 20 h; 50, 100, and 150 µM per 4 h; 150 µM for 16 h	Not present	↓ SOD1 promoter activity at 100 µM for 4 h and 20 h, and at 150 µM for 4 h	↓ HRE-mediated gene transcription at 150 µM for 16 h
[35]	Mouse skeletal muscle cell line (C2C12)	100 µM for 24 h	Not present	↔ GPx, CAT, SOD1, and SOD2 protein expression; ↓ SOD activity; ↔ GPx and CAT activity	↑ ROS production
[36]	Murine adipocyte cell line (3T3-L1)	Pretreatment with 100 µM for 8 h; 25, 50, and 100 µM for 8 h; 100 µM for 3, 6, 8, and 12 h; 100 µM for 24 h	Subsequent oxidative stress by 0.5 mM for 24 h	Basal condition: ↔ SOD1, SOD2, CAT, and GPx mRNA expression at 100 µM for 8 h	Basal condition: ↑ HO-1 and NQO1 mRNA expression at 100 µM for 8 h; ↑ HO-1 mRNA expression at 50 and 100 µM for 8 h; ↑ protein expression at 100 µM for 24 h; ↔ ROS formation at 100 µM for 1 h
					Oxidative condition: ↓ ROS formation at 100 µM for 1 h
[37]	Human breast carcinoma cell line (MDA-MB-231)	30 µM for 7 d	Concomitant oxidative stress by 5 nM doxorubicin for 7 d	Basal condition: ↓ GPx1 activity and ↑ GSH content; ↔ SOD and CAT activity; ↔ GPx1 mRNA and protein expression	Basal condition: ↔ γGCL and GR activity
				Oxidative conditions: ↓ GPx1 activity and ↑ GSH content; ↔ SOD and CAT activity; ↔ and ↓ GPx1 mRNA and protein expression, respectively	Oxidative condition: ↑ γGCL and GR activity
	Human breast carcinoma cell line (MCF-7)	30 µM for 7 d	Concomitant oxidative stress by 20 nM doxorubicin for 7 d	Basal condition: ↔ SOD, CAT, GPx1 activity, and GSH content	Basal conditions: ↑ and ↔ γGCL and GR activity, respectively
				Oxidative condition: ↔ SOD and CAT activity; ↑ GPx1 activity and ↓ GSH content	Oxidative conditions: ↑ and ↔ γGCL and GR activity, respectively
[38]	Human hepatoma cell line (HepG2)	60 µM DHA for 21 h	Not present	↑ SOD and GPx activity; ↑ GSH content; ↔ CAT activity	↑ GST activity; ↔ TAC and CD content
[39]	Human hepatoma cell line (HepG2)	3.13, 6.25, 12.5, 25, and 50 µM for 24 h	Not present	↑ TAC at 6.25 µM; ↑ GSH content at 25 and 50 µM	↔ HO-1, NQO1, and GSTM2 mRNA expression; ↑ Nrf2 mRNA expression at 3.13 µM; ↑ ARE-mediated gene transcription at 6.25, 12.5, 25, and 50 µM
[40]	Human hepatoma cell line (HepG2)	Pretreatment with 60 µM for 21 h	Subsequent oxidative stress by 0.2 mM H_2_O_2_ for 20 min and 1 h	20 min oxidative stress: ↔ SOD, CAT, and GPx activity	20 min oxidative stress: ↔ GST activity
				1 h oxidative stress: ↑ SOD activity; ↑ TAC and GSH content; ↔ GPx and CAT activity	1 h oxidative stress: ↔ GST activity; ↔ TBARS and CD content
[41]	Human hepatoma cell line (HepG2)	50 and 100 µM for 48 h	Not present	↑ TAC; ↑ FRAP and GPx activity at 100 µM; ↑ SOD and CAT activity; ↑ SOD2 and GPx1 mRNA expression at 100 µM; ↑ CAT mRNA expression	↓ mitochondrial ROS; ↓ ROS level at 100 µM; ↑ ADP/ATP ratio and ↓ ATP level at 100 µM; ↑ MMP level; ↔ ND1, ND5, and Nrf2; ↑ COX1, COX3, TFAM, TFB1M, TFB2M, Nrf1, ERRα, and PGC1α mRNA expression at 100 µM
[42]	Human hepatoma cell line (HepG2)	Pretreatment with 50 µM for 24 h	Subsequent oxidative stress by 6 mM H_2_O_2_ for 1 h	Oxidative condition: ↑ CAT and GPx activity; ↔ SOD activity	Oxidative condition: ↓ MDA content
[43]	Mouse hepatocyte cell line (AML12)	Pretreatment with 50 µM for 12 h	Subsequent oxidative stress by 0.4 mM H_2_O_2_ for 2 h	Oxidative condition: ↑ SOD1 protein expression	Oxidative condition: ↑ HO-1, total and mitochondrial LC3BI/II, beclin1, parkin, PINK1, PGC1α, and p-ERK/ERK protein expression; ↓ P62/SQSTM2 and Drp1 protein expression; ↔ p-AMPK/AMPK; ↓ ROS content and apoptosis; ↑ MMP
[44]	Mouse hippocampal nerve cell line (HT22)	17.2 µM for 48 h and 54 h	Not present	↔ SOD1, SOD2, CAT, GPX1, and PRDX5 mRNA expression at 48 h; ↑ PRDX2 and PRDX3 mRNA expression at 48 h; ↓ PRDX4 mRNA expression at 48 h; ↑ GPx activity; ↑ total GS and GSH at 54 h	↔ TXNIP, TXNRD3, GLRX1, m/c GLRX2, and NXN mRNA expression at 48 h; ↑ TXN1, TXN2, TXNRD1, TXNRD2, GCLC, GR, mGPx4, m/cGPx4, nGPx4, and SRXN1 mRNA expression at 48 h; ↔ GST activity; ↑ TBARS content; ↑ GR and GPx4 activity; ↑ TrxR at 54 h; ↑ cell DHA content at 48 h
[45]	Rat glioblastoma cell line (C6)	25, 50, and 75 µM for 24, 48, and 72 h	Not present	↔ GSH content for 24 h; ↑ CAT activity at 50 µM and 75 µM for 48 h; ↑ GPx activity at 25 µM for 24 h and 48 h; ↑ and ↓ GPx activity at 50 µM for 24 h and 75 µM for 48 h, respectively	↑ cell DHA and ROS content; ↓ TBARS content at 24 h and ↑ at 72 h; ↑ TBARS content at 75 µM for 48 h; ↔ G6PDH for 48 h
[46]	Human neuroblastoma cell line (SH-SY5Y)	Pretreatment with 0.01, 0.1, and 1 µM for 24 h	Subsequent oxidative stress by 3 µM MeHg for 2 h	Basal condition: ↔ GPx1 mRNA expression; ↑ CAT and SOD1 mRNA expression at 1 µM	Oxidative condition: ↓ ROS content
[47]	Mouse Schwann cell line (IMS32)	Pretreatment with 7.5 and 12.5 µM for 4 h; 7.5 µM for 3 h, 6 h, 12 h, and 24 h	Subsequent oxidative stress by 50 µM t-BHT for 1.5 h	Basal condition: ↑ CAT mRNA expression at 7.5 and 12.5 µM for 4 h; ↔ SOD and GPx mRNA expression at 7.5 and 12.5 µM for 4 h; ↑ CAT mRNA expression at 6 and 12 h; ↑ CAT activity at 12 h; ↑ GSH and ↔ GSSG content at 12 h	Basal condition: ↑ HO-1 and NQO1 mRNA expression at 7.5 and 12.5 µM for 4 h; ↑ HO-1 mRNA expression at 3, 6, and 12 h; ↑ NQO1 mRNA expression at 6, 12, and 24 h; ↑ HO-1 protein expression at 6, 12, and 24 h; ↑ NQO1 protein expression at 12 and 24 h; ↑ ARE activity at 12 h; ↑ and ↓ nuclear and cytoplasmic Nrf2 protein expression at 3 and 6 h, respectively
					Oxidative condition: ↓ ROS production
[48]	Rat pheochromocytoma cell line (PC12)	Pretreatment with 40 µg/mL PS-containing DHA for 24 h	Subsequent oxidative stress by 0.2 mM H_2_O_2_ and 0.15 mM t-BHT for 4 h	H_2_O_2_ oxidative condition: ↑ TAC and SOD activity	t-BHT oxidative condition: ↓ caspase 9 and caspase 3 mRNA expression; ↔ akt-2 and GSK-3β mRNA expression; ↑ bcl-2 mRNA expression; ↓ bax and ↑ bcl-2 protein expression
				t-BHT oxidative condition: ↑ TAC and SOD activity	
[49]	Rat pheochromocytoma cell line (PC12)	Pretreatment with 12.5 µM for 4 h	Subsequent stress by 50 µM Aβ_25−35_ for 48 h	Cytotoxic condition: ↑ TAC; ↔ GSH content and GSSG/GSH ratio	Cytotoxic condition: ↓ MDA and ROS content; ↑ nucleus/cytoplasmic Nrf2 ratio; ↔ γGCLC mRNA and protein expression; ↓ and ↔ γGCLM mRNA and protein expression, respectively; ↔ and ↑ HO-1 mRNA and protein expression, respectively; ↑ and ↔ NQO1mRNA and protein expression, respectively; ↓ CD36 mRNA and protein expression; ↓ SRB1 and FABP5 mRNA expression; ↔ SRB1 and FABP5 protein expression
[50]	Rat pheochromocytoma cell line (PC12)	Pretreatment with 60 µM for 24 h	Subsequent oxidative stress by 0.3 mM H_2_O_2_ for 24 h	Basal condition: ↑ SOD and GPx activity; ↔ CAT activity; ↑ GSH content	Basal: ↔ ROS, MDA, ascorbic acid, nitrate and nitrite content; ↔ MMP; ↔ bax and bcl2 mRNA expression and protein content; ↔ Nrf2 and HO-1 mRNA expression; ↑ Nrf2 and HO-1 protein expression
				Oxidative condition: ↑ SOD and GPx activity; ↔ CAT activity; ↑ GSH content	Oxidative condition: ↓ ROS and MDA content; ↔ nitrate, nitrite, and ascorbic acid content; ↑ MMP; ↓ bax mRNA expression and protein content; ↑ bcl2 mRNA expression and protein content; ↑ Nrf2 and HO-1 mRNA and protein expression
[51]	Rat pheochromocytoma cell line (PC12)	Pretreatment with 20, 40, and 80 µM for 24 h	Subsequent oxidative stress by 0.4 mM H_2_O_2_ for 12 h	Oxidative condition: ↑ SOD activity	Oxidative condition: ↓ ROS and MDA content; ↑ Na^+^/K^+^-ATPase and ↔ Ca^2+^/Mg^2+^-ATPase; ↑ p-TrkB, p-PLCγ1, p-CaMKII, p-ERK 1/2, and p-CREB protein expression; ↓ AchE and ↑ ChAT activity; ↑ Ach and GABA content; ↓ Glu content
[52]	Rat pancreatic ß cell line (INS-1E)	25, 50, 75, and 100 µM for 48 h	Not present	↔ SOD1 and SOD2 protein expression; ↑ GPx1 protein expression	↔ TXN1, GPR40, glycosylated, and unglycosilated gp 91 protein expression; ↓ and ↔ ROS content at 100 µM and NO content, respectively; ↔ secreted and ↑ intracellular insulin content
[53]	Rat pancreatic acinar cell line (AR42J)	Pretreatment with 20 and 50 µM for 2 h	Subsequent inflammatory/oxidative stress by 0.1 µM cerulein for 1, 4, and 24 h	Basal condition: ↔ CAT mRNA and protein expression	Basal condition: ↔ IL-6 mRNA and p-JAK2, p-STAT3, nPPARγ, and cPPARγ protein expression; ↔ IL-6 and ROS content
				Inflammatory/oxidative condition: ↑ CAT mRNA and protein expression	Inflammatory/oxidative condition: ↓ IL-6 mRNA and p-JAK2, p-STAT3, and cPPARγ protein expression; ↓ IL-6 and ROS content; ↑ nPPARγ protein expression
[54]	Human monocytic cell line (THP-1)	10, 25, 50, and 100 µM for 24 h	Concomitant inflammatory stress by 100 µg/mL (0.53 mM) MSU for 24 h	Basal condition: ↔ SOD protein expression at 50 µM	Basal condition: ↔ NLRP3 mRNA expression and NLRP3, caspase-1, and IL-1β precursor protein expression at 50 µM; ↓ mitochondrial ROS content at 50 µM; ↑ TXNIP, HO-1, and NQO1 protein expression at 50 µM; ↔ total, ↑ nuclear, and ↓ cytosolic Nrf2 protein expression at 50 µM
				Inflammatory condition: ↓, ↔, ↑, SOD, GPx, and CAT mRNA expression, respectively; ↔ SOD protein expression at 50 µM	Inflammatory condition: ↓ IL-1β and TNF-α mRNA and protein expression; ↓ and ↔ NLRP3 mRNA and protein expression at 50 µM, respectively; ↓ IL-1β precursor and ↔ caspase-1 protein expression at 50 µM; ↓ ROS content and mitochondrial ROS content at 50 µM; ↑ HO-1 and NQO1 mRNA expression; ↑ TXNIP, HO-1, and NQO1 protein expression at 50 µM; ↔ total, ↑ nuclear, and ↓ cytosolic Nrf2 protein expression at 50 µM

Effects refer to respective unsupplemented control cells either in basal or stressed conditions. When DHA concentrations/times are not reported, the effects refer to all experimental conditions tested. Abbreviations: ↑: increase; ↓: decrease; ↔: no effect; 8-OHdG: 8-hydroxy-2-deoxyguanosine; MeHg: metilmercury; Ach: acetylcholine; AchE: acetylcholinesterase; ADP: adenosine diphosphate; Akt-2: serine/threonine kinase 2; AMP: adenosine monophosphate; AMPK: phospho-AMP-activated protein kinase; ARE: antioxidant response element; ATP: adenosine triphosphate; Aβ_25−35_: amyloid beta-peptide 25-35; Bax: Bcl-2-associated X protein; Bcl-2: B-cell lymphoma 2; cAMP: cyclic adenosine monophosphate; CAT: catalase; CD: conjugated dienes; CD36: cluster of differentiation 36; cGMP: cyclic guanosine monophosphate; cGPx: cytosolic glutathione peroxidase; ChAT: choline acetyltransferase; COX: cyclooxygenase; COX1: cytochrome c oxidase I; COX3: cytochrome c oxidase subunit III; cPPARγ: cytosolic peroxisome proliferator-activated receptor gamma; Cys: cysteine; DHA: docosahexaenoic acid; Drp1: dynamin-related protein 1; ERK: extracellular signal-regulated kinase; ERRα: estrogen-related receptor α; FABP5: fatty acid binding protein 5; FRAP: ferric reducing antioxidant power; G6PDH: glucose-6-phosphate dehydrogenase; GABA: gamma-aminobutyric acid; GCLC: glutamate cysteine ligase, catalytic subunit; GLRX1; glutaredoxin 1; Glu: glutamic acid; GPR40: G-protein-coupled receptor 40; GPx: glutathione peroxidase; GPx1: glutathione peroxidase 1; GPx4: phospholipid-hydroperoxide glutathione peroxidase 4; GR: glutathione reductase; GS: glutathione; GSH: reduced glutathione; GSK-3β: glycogen synthase kinase 3 beta; GSSG: glutathione oxidized; GST: glutathione S-transferase; GSTM2: glutathione S-transferase mu 2; H_2_O_2_: hydrogen peroxide; HO-1: heme oxygenase-1; HRE: hypoxia response element; IL-1β: interleukin-1β; LC3BI/II: microtubule-associated protein 1 light chain 3 isoform B; m/cGLRX2: mithocondrial/cytoplasmic glutaredoxin 2; m/cGPx4: mithocondrial/cytosolic phospholipid-hydroperoxide glutathione peroxidase 4; MDA: malondialdehyde; mGPx4: mithocondrial phospholipid-hydroperoxide glutathione peroxidase 4; MMP: mitochondrial membrane potential; MSU: monosodium urate; NADH: nicotinamide adenine dinucleotide; ND1: NADH-ubiquinone oxidoreductase chain 1; ND5: NADH-ubiquinone oxidoreductase chain 5; nGPx4: nuclear phospholipid-hydroperoxide glutathione peroxidase 4; NLRP3: NLR family pyrin domain containing 3; NO: nitric oxide; nPPARγ: nuclear peroxisome proliferator-activated receptor gamma; NQO1: NAD(P)H quinone dehydrogenase 1; Nrf1: nuclear respiratory factor 1; Nrf2: nuclear factor (erythroid-derived 2)-like 2; NXN: nucleoredoxin; P62: ubiquitin-binding protein p62; p-AMPK: phospho-AMP-activated protein kinase; PC: phosphatidylcholine; p-CaMKII: phospho-Ca^2+^/calmodulin-dependent protein kinase II; p-CREB: phospho-cyclic adenosine monophosphate response element-binding protein; PDE: phosphodiesterase; PE: phosphatidyletanolamine; p-ERK 1/2: phospho-extracellular signal-regulated kinases 1/2; p-ERK: phosphor-extracellular signal-regulated kinase; PGC1α: peroxisome proliferator-activated receptor gamma coactivator 1α; PINK1: phosphatase and tensin homolog-induced kinase 1; p-JAK2: phospho-janus kinase 2; PL: phospholipid; p-PLCγ1: phospho-phospholipase C, gamma 1; PRDX2: peroxiredoxin 2; PRDX3: peroxiredoxin 3; PRDX4: peroxiredoxin 4; PRDX5: peroxiredoxin 5; PS: phosphatidylserine; p-STAT3: phospho-signal transducer and activator of transcription 3; ROS: reactive oxygen species; SOD: superoxide dismutase; SOD1: superoxide dismutase 1; SOD2: superoxide dismutase 2; SQSTM2: sequestosome-2; SRB1: scavenger receptor, class B type 1; SRXN1: sulforedoxin 1; TAC: total antioxidant capacity; TBARS: thiobarbituric acid reactive substances; t-BHT: tert-butylhydroperoxide; TCDD: 2,3,7,8-tetrachlorodibenzo-p-dioxin; TFAM: mitochondrial transcription factor A; TFB1M: dimethyladenosine transferase 1, mitochondrial; TFB2M: dimethyladenosine transferase 2, mitochondrial; TNF-α: tumor necrosis factor-α; TOS: total oxidative stress; TrkB: tropomyosin receptor kinase B; TXN1: thioredoxin 1; TXN2: thioredoxin 2; TXNIP: thioredoxin-interacting protein; TXNRD1: thioredoxin reductase 1; TXNRD2: thioredoxin reductase 2; TXNRD3: thioredoxin reductase 3; UCB: unconjugated bilirubin; α-TC: α-tocopherol; γGCL: γ-glutamate cysteine ligase; γGCLC: γ-glutamate cysteine ligase catalytic subunit; γGCLM: γ-glutamate cysteine ligase modifier subunit; and γTC: γ-tocopherol.

**Table 2 antioxidants-12-01283-t002:** Summary of findings related to the effect of DHA feeding in animal models.

Ref.	Animal Model	DHA Somministration	Treatment	Primary Outcomes	Secondary Metabolic Outcomes
[55]	Male Wistar rats	250 and 1500 mg/d/kg bw for 5 d by gastric intubation	Not present	250 mg/d/kg bw: ↔ plasmatic and hepatic GSH content	250 mg/d/kg bw: ↔ plasmatic VE, VA, MDA, Cys, and Cys-Gly content; ↔ hepatic VE, VA, MDA, Cys, and HC content; ↔ hepatic XOX, XDH, and ACO activity
				1500 mg/d/kg bw: ↔ hepatic SOD1, SOD2 mRNA expression; hepatic GPx and SOD activity; ↔ plasmatic and hepatic GSH	1500 mg/d/kg bw: ↔ plasmatic VA, Cys, Cys-Gly, and HC; ↑ and ↓ plasmatic MDA and VE; ↔ hepatic VE, VA, AsA, and Cys content; ↑ hepatic MDA content and ACO activity; ↔ hepatic XOX, GR, GST, and XDH activity
[56]	Male AsA-requiring ODS rats	DHA-rich diet (5 g/kg diet), *ad libitum* for 12 h/d for 32 d	Not present	↑ and ↔ hepatic and renal GSH content, respectively	↑ hepatic CD, TBARS, and DHA content; ↓ hepatic α-TC and AsA content; ↑ renal TBARS and DHA content; ↓ renal AsA content; ↔ testicular TBARS level and AsA content; ↑ testicular DHA content
[57]	Male Wister rats	13.3 mg/d/kg bw for 12w by oral gavage	STZ-induced diabetes (single dose at 65 mg/kg bw)	Normal rats: ↔ retinal GSH content and GPx activity	Normal rats: ↔ retinal MDA and nitrotyrosine content; ↔ total retinal, INL, and ONL thickness; ↔ n° of cells in retinal GC layer, n° of row in retinal ONL, and n° of row in retinal INL; ↔ apoptosis in retinal cells
				Diabetic rats: ↑ retinal GSH content and GPx activity	Diabetic rats: ↔ and ↓ retinal nitrotyrosine and MDA content, respectively; ↔ total retinal, INL, and ONL thickness; ↔ n° of ganglion cells, n° of row in ONL, and n° of row in INL; ↓ apoptosis in retinal cells
[58]	Male Wistar rats	13.3 mg/d/kg bw for 12w by oral gavage	STZ-induced diabetes (single dose at 65 mg/kg bw)	Normal rats: ↔ cortexal GSH content and GPx activity	Normal rats: ↔ cortexal MDA content and n° of 4-HNE positive cells
				Diabetic rats: ↑ cortexal GSH content and GPx activity	Diabetic rats: ↓ cortexal MDA content and n° of 4-HNE positive cells
[59]	Sprague-Dawley rats	2.5 mg/d/kg bw for 36 d by oral gavage	PTZ-induced epilepsy (repetitive doses ranging from 45 to 10 mg/kg bw)	Epileptic rats: ↑ cerebral SOD and CAT activity; ↑ and ↔ hepatic CAT and SOD activity, respectively	Epileptic rats: ↓ and ↔ hepatic and cerebral NO content, respectively
[60]	Sprague-Dawley rats	Intraperitoneally 250 mg/d/kg bw for 21 d	TCDD-induced toxicity (8 µg/d/kg bw for 21 d)	Normal rats: ↑ hepatic GSH content and SOD, CAT, and GPx activity	Normal rats: ↔ MNHEP
				Stressed rats: ↑ hepatic GSH content and SOD, CAT, and GPx activity	Stressed rats: ↓ MNHEP
[61]	Sprague-Dawley rats	2.5 mg/d/kg bw for 36 d by oral gavage	Not present	↑ cerebral CAT and SOD activity; ↑ hepatic GPx and SOD activity; ↔ hepatic CAT and cerebral GPx activity	↔ cerebral and hepatic NO content
[62]	Male Wistar rats	100 mg/d/kg bw for 90 d by oral gavage	AlCl_3_-induced neurotoxicity (100 mg/d/kg bw for 90 d)	Normal rats: ↔ cerebral SOD, CAT, and GPx activity	Normal rats: ↔ cerebral GR activity; ↔ cerebral lipoperoxides and protein carbonyl content
				Stressed rats: ↑ cerebral SOD, CAT, and GPx activity	Stressed rats: ↔ cerebral GR activity; ↓ cerebral lipoperoxides and protein carbonyl content
[63]	Male offspring Wistar rats	75, 150, and 300 mg/d/kg for 21 d by oral gavage	VPA-induced autism (single dose to pregnant rats at 600 mg/kg bw)	Autistic rats: ↑ hippocampal GSH content and SOD and GPx activity at 150 and 300 mg/d/kg bw	Autistic rats: ↓ hippocampal MDA content at 150 and 300 mg/d/kg bw; ↓ hippocampal caspase-3 protein expression and activity, and TUNEL-positive cells at 150 and 300 mg/d/kg bw; ↑ hippocampal BDNF, p-Akt/Akt, p-CaMKII/CaMKII, and p-CREB/CREB protein expression at 150 and 300 mg/d/kg bw; ↑ and ↓ hippocampal Bcl-2 and Bax protein expression, respectively; ↑ hippocampal and plasmatic DHA content at 150 and 300 mg/d/kg bw
[64]	Male Wistar rats	Single administration of 555 mg/kg bw by intragastric gavage	Fluid percussion-induced brain traumatic injury	Brain injured rats: ↑ hippocampal SOD and GPx activity	Brain injured rats: ↓ hippocampal MDA content; ↓ hippocampal caspase-3, and Bax protein expression; ↑ and ↓ hippocampal Bcl-2 protein expression and TUNEL-positive cells, respectively; ↑ and ↓ hippocampal nuclear and cytoplasmic Nrf2 protein expression, respectively; ↑ hippocampal HO-1 and NQO1 mRNA and protein expression
[65]	Sprague-Dawley rats	DHA-rich diet (10 g/kg diet) *ad libitum* for 24 d	Not present	↑ plasmatic TAC and erythrocytic SOD activity; ↔ whole blood GPx activity	↔ plasmatic TG, TC, CE, HDL-C, non HDL-C, ASAT, ALAT, MDA, TNF-α, IL-2, and IL-6 content; ↑ DHA content in liver, heart, lung, spleen, kidney, and red blood cells
[66]	Sprague-Dawley newborn rats	300 mg/d/kg bw for 21 d	Pregnant intrauterine growth restriction	Normal newborn rat: ↔ spleenic GPx and CAT mRNA expression	Normal newborn rat: ↔ spleenic IL-4 and INF-γ mRNA expression; ↔ spleenic Bax and Bcl2 protein expression
				Newborn rat from intrauterine growth restricted pregnant: ↑ spleenic GPx and CAT mRNA expression	Newborn rat from intrauterine growth restricted pregnant: ↑ spleenic IL-4 and INF-γ mRNA expression; ↓ and ↑ spleenic Bax and Bcl2 protein expression, respectively
[67]	Male BALB-c mice	Single administration of 100 mg/kg bw by oral gavage	Indomethacin-induced gastric lesion (single dose at 30 mg/kg bw)	Normal mice: ↑ gastric GSH content; ↔ gastric SOD and CAT activity	Normal mice: ↔ gastric MDA, protein carbonyl, MPO, LTB_4_, ICAM-1, RvD1, and NF-kB content; ↓ TNF-α content; ↔ gastric FFA4 protein expression
				Gastric injured mice: ↑ gastric GSH content; ↔ gastric SOD and CAT activity	Gastric injured mice: ↓ gastric MDA, protein carbonyl, MPO, LTB4, ICAM-1, TNF-a, and NF-kB content; ↑ gastric RvD1 content and FFA4 protein expression
[68]	Male C57BL/6J mice	50 mg/d/kg bw for 12 weeks	HFD-induced steatosis (60% fat, 20% protein, and 20% carbohydrate), *ad libitum* for 12 weeks	Normal mice: ↔ hepatic GSH content and GPx, SOD, and CAT activity; ↔ serum TAC	Normal mice: ↔ hepatic SREBP-1c, PPAR-α, NF-kB, and Nrf2 DNA binding activity; ↔ hepatic SREBP-1c, PPAR-α, NF-kB, Nrf2, ACC, FAS, SCD-1, CPT-1α, ACOX-1, TNF-α, IL-6, and IL-1β mRNA expression; ↔ hepatic ACC, FAS, GR, GST, GGT, NQO1, CPT-1α, and ACO activity; ↔ hepatic protein carbonyl, F-8 isoprostane, and TBARS content; ↑ hepatic RvD1 and RvD2 content; ↔ hepatic RvE1 and RvE2; ↔ serum TNF-α, IL-6, and IL-1β content
				Steatotic mice: ↑ hepatic GSH content and GPx, SOD, and CAT activity; ↔ serum TAC	Steatotic mice: ↑ hepatic PPAR-α and Nrf2 DNA binding activity; ↓ hepatic SREBP-1c and NF-kB DNA binding activity; ↑ hepatic PPAR-α, CPT-1α, ACOX-1, and Nrf2 mRNA expression; ↓ hepatic SREBP-1c, ACC, FAS SCD-1, NF-kB, TNF-α, IL-6, and IL-1β mRNA expression; ↑ hepatic GR, GST, GGT, NQO1, CPT-1α, and ACO activity; ↓ hepatic ACC and FAS activity; ↓ hepatic protein carbonyl, F-8 isoprostane, and TBARS content; ↔ hepatic RvE1 and RvE2 content; ↑ hepatic RvD1 and RvD2 content; ↓ serum TNF-α, IL-6, and IL-1β content
[43]	Male C57BL/6J mice	50 mg/d/kg bw for 7 d by intragastric gavage	CCl_4_-induced liver toxicity (single dose at 0.2 mL/kg bw)	Liver injured mice: ↑ hepatic SOD activity and GSH content	Liver injured mice: ↓ serum LDH and ALAT activity; ↔ serum ASAT activity; ↑ hepatic LC3BI/II, beclin1, and P62/SQSTM2 protein expression; ↔ hepatic MDA content

Effects refer to respective unsupplemented or placebo control animals either in normal or injured conditions. When DHA concentrations are not reported, the effects refer to all experimental conditions. In the presence of more than one concentration, only statistically significant changes are reported. Abbreviations: ↑: increase; ↓: decrease; ↔: no effect; 4-HNE: 4-hydroxy-2-nonenal; ACC: acetyl-CoA carboxylase; ACO: acyl-CoA oxidase; ACOX-1: peroxisomal acyl-coenzyme A oxidase 1; Akt: serine/threonine kinase; ALAT: alanine aminotransferase; AsA: ascorbic acid; ASAT: aspartate transaminase; Bax: Bcl-2-associated X protein; Bcl-2: B-cell lymphoma 2; BDNF: brain-derived neurotrophic factor; Bw: body weight; CaMKII: Ca^2+^/calmodulin-dependent protein kinase II; CAT: catalase; CD: conjugated dienes; CE: cholesterol esters; CPT-1α: carnitine-palmitoyl transferase 1α; CREB: cyclic adenosine monophosphate response element-binding protein; Cys: cysteine; Cys-gly: cysteinylglycine; DHA: docosahexaenoic acid; dUTP: deoxyuridine triphosphate; EtOH: ethanol; FAS: fatty acid synthase; FFA4: free fatty acid receptor 4; GGT: γ-glutamyltransferase; GPx: glutathione peroxidase; GPx1: glutathione peroxidase 1; GR: glutathione reductase; GSH: reduced glutathione; GST: glutathione-S-transferase; HC: homocysteine; HDL-C: high density lipoproteins cholesterol; HFD: high-fat diet; HO-1: heme oxygenase-1; ICAM-1: intercellular adhesion molecule 1; IL-1β: interleukin 1β; IL-2: interleukin 2; IL-4: interleukin 4; IL-6: interleukin 6; INF-γ: interferone-γ; INL: inner nuclear layer; LC3BI/II: microtubule-associated protein 1 light chain 3 isoform B; LDH: lactic dehydrogenase; LTB_4_: leukotriene B4; MDA: malondialdehyde; MNHEP: micronucleated hepatocytes; MPO: myeloperoxidase; NF-kB: nuclear factor kappa-light-chain-enhancer of activated B cells; NO: nitric monoxide; NQO1: NAD(P)H quinone dehydrogenase 1; Nrf2: nuclear factor (erythroid-derived 2)-like 2; ODS: osteogenic disorder-Shionogi; ONL: outer nuclear layer; P62: ubiquitin-binding protein p62; p-Akt: phospho-serine/threonine kinase; p-CaMKII: phospho-Ca^2+^/calmodulin-dependent protein kinase II; p-CREB: phosphor-cyclic adenosine monophosphate response element-binding protein; PPAR-α: Peroxisome proliferator-activated receptor α; PTZ: pentylenetetrazol; RvD1: resolvin D1; RvD2: resolvin D2; RvE1: resolvin E1; RvE2: resolvin E2; SCD-1: stearoyl-CoA-desaturase-1; SOD: superoxide dismutase; SOD1: superoxide dismutase1; SOD2: superoxide dismutase2; SQSTM2: sequestosome-2; SREBP-1c: sterol regulatory element-binding transcription factor 1; STZ: streptozotocin; TAC: total antioxidant capacity; TBARS: thiobarbituric acid reactive substances; TC: total cholesterol; TCDD: 2,3,7,8-tetrachlorodibenzo-p-dioxin; TG: triacylglycerol; TNF-α: tumor necrosis factor α; TUNEL: terminal deoxynucleotidyl transferase dUTP nick end labeling; VA: vitamin A; VE: vitamin E; VPA: valproic acid; XDH: xanthine dehydrogenase; XOX: xanthine oxidase; and α-TC: α-tocopherol.

## Data Availability

Not applicable.

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
