# Peer review of "Docosahexaenoic Acid as Master Regulator of Cellular Antioxidant Defenses: A Systematic Review"

_antioxidants, 2023, doi:10.3390/antiox12061283_

Round 1

Reviewer 1 Report

The review by Borgonovi et al. focuses on the positive and encouraging effects of docosapentaenoic acid (DHA) on cellular antioxidant defenses. Despite the relevance of this topic and the review being well-written, I am concerned, and question that the authors only evaluate the positive effects of DHA in redox homeostasis. Selecting/presenting only positive results biases the reader. However, the authors provide an interesting overview of the described effects of DHA on redox biology, highlighting its effect on antioxidant enzymes, oxidative biomarkers, and the molecular mechanisms involved in NRF2 activation. Please find below my comments in a point-by-point manner.

Majors

1 – Abstract – Lines 19 – 20: Why does the review emphasize only positive results? The authors could explore both effects and present an unbiased perspective.

2 - Proofread the manuscript - Some parts of the manuscript can be improved, and some paragraphs need to be more coherent.

3 – Lack of a translational perspective on human diseases, although the review has the purpose of providing molecular mechanisms related to DHA on redox biology, it would be interesting to translate the presented molecular effects from a physiological and pathophysiological perspective. Why did the authors not mention human data/studies? (i.e.: PMID: 26585906; PMID: 26237736; PMID: 32653511; PMID: 35341783; PMID: 15912889)

Minor points:

4 – Please, double-check the table legend (i.e., lines 159-160).

5 – Please, double-check Figure 2: There is one NRF2 molecule that misses the phosphorylation marker. Also, the arrow from KEAP1, after the dissociation of NRF2, indicating NRF2 going to the nucleus, is confusing. You could indicate the dissociation before, add the pathway where KEAP1 will go without NRF2, and present NRF2 in the nucleus. Another point is about KEAP1 after the dissociation of NRF2: In one of the molecules, it was indicated this dissociation kept the SH group on the KEAP1, you should indicate the sulfhydryl ligament (S-S).

Reviewer 2 Report

This review manuscript focuses on docosahexaenoic acid (DHA) in regulating cellular antioxidant defense. The antioxidant activity of DHA has been extensively studied, and, in this view, a systematic
thorough review would be of great interest to the readers to quickly pick up the major findings in the field. In brief, the authors collected DHA-related works, excluded un-related ones, and narrowed down to 43 publications for analysis. The 43 publications were further grouped to cell culture and animal studies. Overall, the study was well-designed and properly conducted. However, the summaries of findings are relatively descriptive and lacking key points. Below are detailed comments for the authors to consider.

1   1. It is highly suggested to summarize the major findings of the retrieved studies. For example, what were the major findings/conclusions of the cell culture studies? Was there an origin effect of cell type differences deciding whether DHA is protective or toxic? What seems to be a safe dosage range? … These are just some examples. In the current format, there are only over 300 words describing findings of the cell cultures studies and 200 words for animal studies. Also, it is descriptive.

2.      Studies used DHA oxidation products were excluded from the review, I am not sure whether it was a good choice. In the manuscript, it mentioned that DHA metabolites, rather than DHA itself, serve as activators of Nrf2 and inducers of antioxidative enzymes.

    2. A follow-up question to the previous point is the inclusion rate of the topic. For example, PMID 33497204 was not included in the review, but it seems meet all the criteria.

     3. The Abstract. Only the last 2 sentences were about the manuscript, whereas the major text was about background information. Please revise it.

    4. The main text is about 2600 words, which is shorter than a typical review. It is suggested to extend the contents by, for example, adding key points of the studies as discussed in comment 1.     

Author Response

This review manuscript focuses on docosahexaenoic acid (DHA) in regulating cellular antioxidant defense. The antioxidant activity of DHA has been extensively studied, and, in this view, a systematic
thorough review would be of great interest to the readers to quickly pick up the major findings in the field. In brief, the authors collected DHA-related works, excluded un-related ones, and narrowed down to 43 publications for analysis. The 43 publications were further grouped to cell culture and animal studies. Overall, the study was well-designed and properly conducted. However, the summaries of findings are relatively descriptive and lacking key points. Below are detailed comments for the authors to consider.

We thank the reviewer for her/his appreciation of our work. The manuscript has been improved following the reviewer's suggestions (all modifications are in red) and we now believe it has reached sufficient quality to be published in Antioxidants

1   1. It is highly suggested to summarize the major findings of the retrieved studies. For example, what were the major findings/conclusions of the cell culture studies? Was there an origin effect of cell type differences deciding whether DHA is protective or toxic? What seems to be a safe dosage range? … These are just some examples. In the current format, there are only over 300 words describing findings of the cell cultures studies and 200 words for animal studies. Also, it is descriptive.

We thank the reviewer for her/his comment but, as we indicated in the conclusion, although most of the selected studies demonstrated an encouraging effect of DHA in modulating antioxidant defenses, the different experimental conditions adopted such as cell/animal models, timing of supplementation/treatment, concentrations used, exogenous stresses, tissues investigated, and methodologies adopted among the various studies do not allow us to clearly establish a specific efficacy value, safe dose, or minimum treatment time to achieve such positive effects. For example, DHA supplemented at the 30 micromolar concentration for 48 hours resulted in increased GSH content in human fibroblasts (ref 27 in the revised versions), while the same concentration for 7 days resulted in no change in the content of this tripeptide in human breast carcinoma cell line (ref 36 in the revised version). Precisely for these reasons in the conclusions of the manuscript we have further indicated the importance of carrying out further studies to evaluate the minimum effective treatment times and concentrations of DHA according to the cellular models adopted. We are sorry, but we do not believe there is enough data to confidently extrapolate the information indicated by the reviewer, whom we thank for the suggestion anyway. We also totally agree with the reviewer's suggestion to implement the description of the results. Therefore, we have included in the results section the description of the change in oxidation markers as well as whether the assessment of antioxidant defenses was evaluated at the transcriptional, posttranscriptional, or enzyme activity level.

  1. Studies used DHA oxidation products were excluded from the review, I am not sure whether it was a good choice. In the manuscript, it mentioned that DHA metabolites, rather than DHA itself, serve as activators of Nrf2 and inducers of antioxidative enzymes.

We thank the Reviewer for her/his comment. After careful discussion among the authors, it was decided not to include within the research strategy studies performed using DHA oxidation products for a number of reasons. The first reason lies in the fact that DHA is normally taken in through the diet in a non-oxidized form therefore the use of DHA oxides may not fully reflect the events that occur from a nutritional point of view. Secondly, previous studies have demonstrated that cultured cells such as endothelial cells, smooth muscle cells, and macrophages take up oxidized fatty acids poorly as compared with unoxidized fatty acids (10.1016/S0022-2275(20)33460-X). Third, DHA is not necessarily oxidized once it is incorporated within cellular lipids. In fact, in vitro and in vivo studies have shown how there is no increase in oxidation products (ref 28, 37, 39, 49, 55-57, 61, 66, 67 in the revised version) or even a decrease in them (ref 25, 28, 30, 41, 44, 48, 50, 61-63, 67 in the revised version). In this sense, DHA may act, as discussed in the review, possibly by acting as an activator of Nrf2 through phosphorylation pathways via activation of cellular kinases or as ligands of other nuclear receptors such as PPARs. Last, with this review we wanted offer a vision providing molecular-based support to any recommendation pertaining the food/health relationship, where the most recent recommendations exclude to introduce oxidized fatty acid with the diet. In any case we have better clarified thie reason of this exclusion criteria in the methods section

  1. A follow-up question to the previous point is the inclusion rate of the topic. For example, PMID 33497204 was not included in the review, but it seems meet all the criteria.

We thank the Reviewer for her/his careful analysis but the article indicated had been included in the review (ref 37 in the original version and ref 40 in the revised version)

  1. The Abstract. Only the last 2 sentences were about the manuscript, whereas the major text was about background information. Please revise it.

According to the Reviewer’s suggestion, the abstract has been improved.

  1. The main text is about 2600 words, which is shorter than a typical review. It is suggested to extend the contents by, for example, adding key points of the studies as discussed in comment 1.

We thank the Reviewer for her/his honorable comment. The manuscript has been expanded and some important points in discussion and conclusion sections have been more commented and described. We have also extended the introduction to include the cellular mechanisms of DHA uptake and the effects of increasing lipid unsaturation at the level of cell membranes.

Reviewer 3 Report

Authors summarize effects of Docosahexaenoic acid (DHA) on cellular redox status discussing its potential beneficial properties. This study could be very valuable for readers from broad fields. However some clarifications, explanation of concepts and clear definitions of consensus findings and speculations must be given.

·       In the abstract the statement  “DHA might maintain the cellular redox status while encouraging the transcriptional regulation of antioxidants by Nrf2 activation” is not entirely clear. Please explain it better.

·       Specify at least a handful of Nrf2 target genes. Moreover authors should explain which genes are modulated directly by DHA and how this is elicited. Is DHA binding to GPCRs at all?

·       In the introduction is not clear if authors are proposing a relationship between DHA and Nrf2 or this connection is already established.

·       Authors should talk more on the variation in the dose given in studies in vitro and in vivo and what is reported in the literature behind a dose effect for DHA. Moreover, authors should give more details about what they talk in the discussion regarding cell differences in taking up DHA, if all the cells have the machineries to put DHA in the membranes, and also mention in vivo PK/PD studies on DHA.

·       The conclusions for paragraph 3.1 are not clear regarding authors mentioning “one” promoting effect. Authors should provide an organic summary for this paragraph with a consensus effect that could be derived by the analysis of the studies involved. Finally authors should speculate on the positive vs negative effects seen in the research studies.

·       Authors should briefly describe what “primary outcomes” and “ secondary metabolic outcomes” mean regarding the tables.

·       Authors should discuss what happens when a cell has a lot of unsaturated fats in terms of redox capacity and metabolism.

·       The phrase “susceptibility to oxidation by ROS is not even as straightforward as hypothesized from a theoretical viewpoint” is not clear.

·       Please clarify what are the “ligands” for Nrf2 and what experiments have been done to rule out DHA as a ligand in this setting.

·       Is ref 80 the only indication of a direct activation of Nrf2 by DHA?

·       At some point it could seem that authors believe that the effect of DHA on Nrf2 is only indirect via the radicals downstream of DHA. Authors should make some conclusions on the data highlighted in the table, where the "secondary outcome" many time involves upregulation of Nrf2 target genes, while some studies also have looked at Nrf2 nuclear translocation. However, later the authors switch to a view where s DHA acts on Nrf2 dually, via ROS-KEAP mechanism and via lipid composition of membranes-MAPK-phosphoNrf2. It appears that the phospho-Nrf2 outcome is not present in the studies authors mention in the table: as anybody looked at p-Nrf2 after DHA treatment? Authors should provide a more definite conclusion on the data and separate well concepts from speculations.

·       Please briefly describe the molecular mechanism by which DHA is incorporated in the plasma membrane.

n/a

Round 2

Reviewer 2 Report

No further comments. 

Author Response

Thank you

Reviewer 3 Report

Authors have addressed nicely the majority of reviewers comments.

One more clarification must be given before publication concerning 2 important aspects. It is indeed not clear the role of DHA in membrane fluidity and in the fact that being unsaturated it is a major target of ROS. How do these 2 aspects support its antioxidant role? Does its unsaturation helps antioxidant capacity because it is preventing lipid oxidation of other cellular important lipids? So the role of DHA would be to just self inflict damage preventing broad cellular damage? About the fluidity and membrane composition, authors mention GPCR activity in this scenario: how does the activity of these receptors change in response to DHA molecules being part of membrane bilayer and what are the antioxidants benefits as a result?

n/a

Round 3

Reviewer 3 Report

the mancuscript has been improved with the last clarifications.